



# The Applicability of the Integral Method with Variable Limit in Solving the Governing Equations for Temperature and Salinity in an Ocean Circulation Model

Xiaole li[1,2], Zhenya Song[1], Xiongbo Zheng[2], Zhanpeng Zhuang[1], Fangli Qiao[1], Haibin Zhou[2], and Mingze Ji[2]

[1]First Institute of Oceanography Ministry of Natural Resources, Qingdao, China
[2]College of Mathematical Sciences, Harbin Engineering University, Harbin, China

**Correspondence:** Fangli Qiao (qiaofl@fio.org.cn)

**Abstract.** To address limitations in traditional discretization methods for ocean numerical modeling, this study develops a integral method with variable limit (IMVL) to enhance the simulation accuracy of thermohaline dynamics in ocean models. Under the Arakawa-C grid framework, we propose novel discretization schemes applying variable-limit integration to horizontal advection, horizontal diffusion, and vertical diffusion terms in temperature-salinity equations. For the vertical diffusion

term, the variable limit integral scheme is also designed and combined with the time discretization of the difference method to form an implicit fully discrete scheme. Stability analysis based on convection equation principles confirms the numerical robustness of the proposed method. Implementation within the Princeton Ocean Model (POM) demonstrates significant improvements: 1) Strait test cases reveal 40-60% error reduction in temperature-salinity simulations compared to standard POM; 2) Enhanced topographic sensitivity enables superior representation of overflow dynamics across steep sills; 3) The modified

scheme eliminates numerical instabilities in zero-Coriolis scenarios, maintaining physical validity beyond 720 simulation days by preventing artificial water stacking and gradient accumulation. The computational efficiency analysis demonstrates that the introduction of the variable-bound integration method increases the total runtime by merely 25%. These findings establish the variable-limit integration method as an effective approach for improving the dynamic framework of ocean models, particularly demonstrating outstanding performance in enhancing model stability and resolving dynamic processes under complex

topographies. It is noteworthy that the variable-limit integral method designed herein for the thermohaline equations represents a novel and more stable solution approach, which, while implemented in POM within this study, is universally applicable to other ocean numerical models.

Keywords: Ocean numerical model; Variable limit integral method; Thermohaline equation

## 1 Introduction

The ocean numerical model can quantitatively describe marine physical phenomena and their dynamic processes, serving as a critical tool for large-scale, comprehensive, and sustained oceanographic research (Feng et al., 1999). This model has been



extensively utilized in simulating and predicting climate and environmental changes, while also providing technical support for marine fisheries, shipping, marine engineering construction, and resource development (Zhao et al., 2014).

Ocean numerical models simulate seawater temperature and salinity fields, along with density distributions, by numerically
solving the governing equations of oceanic motion. These equations encompass highly complex physical processes involving dynamic and thermodynamic interactions within the marine environment. Due to their nonlinear and multiscale nature, closed-form analytical solutions to these governing equations are generally intractable. Consequently, the accuracy of ocean numerical simulations fundamentally relies on the development of robust numerical methods that ensure high precision, numerical stability, and computational efficiency in solving the governing equation system.

Current marine numerical models predominantly employ finite difference and finite volume methods for discretization. Widely adopted finite difference-based models include the Princeton Ocean Model (POM) (Mellor et al., 1994; Ezer et al., 2002), the Hybrid Coordinate Ocean Model (HYCOM) (Bleck, 2002; Halliwell, 2004), the Regional Ocean Modeling System (ROMS) (Shchepetkin and McWilliams, 2005; Song and Haidvogel, 1994; Shchepetkin and McWilliams, 2009), and the MAS-NUM three-dimensional ocean model (Han, 2014; Han and Yuan, 2014). These schemes typically utilize central difference or
upwind formulations for spatial discretization, coupled with Euler or leapfrog schemes for temporal integration.Theoretically, most mainstream finite difference-based marine models achieve first-order accuracy on non-uniform grids and second-order accuracy on uniform grids. However, such approaches frequently require empirical adjustments to maintain numerical stability, particularly when handling sharp gradients or complex bathymetric features. This inherent trade-off between accuracy and stability underscores the need for advanced numerical frameworks in modern ocean modeling.

Several ocean numerical models utilizing the finite volume method have been developed, including the MIT General Circulation Model (MITgcm) (Adcroft and Campin, 2004; Marotzke et al., 1999), widely applied in coupled atmosphere-ocean circulation studies; the Finite Volume Community Ocean Model (FVCOM) (Chen et al., 2006), integrating hydrodynamic and ecological modules; and the unstructured grid-based three-dimensional framework (Casulli and Walters, 2000). These models typically employ triangular or unstructured meshes to resolve complex coastal geometries, with some incorporating
conservative numerical schemes to ensure mass-energy conservation. However, higher-order finite volume implementations often necessitate specialized techniques to mitigate numerical dispersion (Li et al., 2008) and incur substantial computational costs, particularly in multiscale simulations, highlighting the persistent challenge of balancing accuracy with computational feasibility in modern ocean modeling.

Recent advancements in numerical methodologies have substantially advanced the simulation of ocean dynamics. A pivotal
development emerged in 2020 when Subich et al. successfully adapted the semi-Lagrangian advection scheme, previously utilized in atmospheric modeling, to the NEMO ocean model. This adaptation overcame the time-step limitations imposed by the Courant-Friedrichs-Lewy (CFL) condition in high-resolution configurations (Subich et al., 2020). Building on this progress, Lan et al. (2022) introduced a dual-rate explicit time integration framework within the Runge-Kutta method, specifically designed to resolve the multiscale interactions between baroclinic and barotropic modes. These innovations collectively enhance
the numerical precision and computational efficiency of ocean models, addressing the inherent stiffness of primitive equation systems while maintaining stability across diverse spatiotemporal scales. In 2022, Patching studied the coarse-graining process




that maintains divergence and gradient in a finite volume primitive equation ocean model (Patching, 2022). In 2023, Boittin et al. developed a low Mach type approximation for a compressible Navier-Stokes system with free-surface flow and temperature and salinity (Boittin et al., 2023). In 2024, Andreas et al. proposed a new semi-Lagrange splitting (SLS) scheme for solving

Euler systems with free surfaces and vertical non-static flows (Alexandris-Galanopoulos et al., 2024).

Recent advances in numerical methods for partial differential equations have seen the emergence of the variable limit integral method, which demonstrates distinct advantages in constructing conservation-preserving, high-order numerical schemes (Luo et al., 2017). Evolved from the finite volume framework and initially applied to wood moisture content calculations (Guo, 2014), this method uniquely performs variable-limit integration near grid nodes. Its defining feature lies in employing undeter-

mined, adaptive integration limits while allowing flexible selection of high-precision approximation functions within localized integration domains. Unlike conventional finite difference approaches that rely on fixed stencil information, the variable limit integral method incorporates weighted contributions from all neighboring points through parametric integration limits. Notably, through strategic parameterization of integration boundaries and approximation functions, the method achieves formal equivalence to finite difference schemes, thereby unifying flexibility in function space discretization with structured grid-based

computational efficiency.

The integral method with variable limit (IMVL) has emerged as a versatile numerical framework for conservative partial differential equations, with demonstrated success across multiple physical systems including the Klein-Gordon equation (Luo et al., 2017), Regularized Long Wave (RLW) equation (Luo et al., 2021), and Rosenau-RLW equation (Guo et al., 2019). Rigorous theoretical analyses establish that these adaptive integral formulations simultaneously achieve two critical objectives:

high-order convergence precision and strict preservation of intrinsic conservation laws. Numerical validations reveal their superior performance characteristics, particularly in long-term simulations where IMVL demonstrates both enhanced accuracy and mitigated error propagation. A representative case study of the Klein-Gordon equation highlights the method's advantages. The fourth-order IMVL implementation achieves 15% greater error reduction compared to conventional fourth-order finite difference schemes at equivalent computational cost. More significantly, longitudinal simulations demonstrate asymptotically slower

error accumulation, with relative accuracy improvements exceeding 20% over 100 characteristic time units. This dual capacity for precision enhancement and computational economy establishes IMVL as a promising candidate for energy-preserving system simulations requiring extended temporal integration.

Comprehensively restructuring the numerical computational framework of ocean models presents a significant challenge. Fortunately, the modular architecture of mainstream systems allows for targeted improvements to the temperature-salinity

equation solver. This study designs new high-precision numerical solution schemes for the temperature and salinity equations using the variable-limit integration method. Based on the Arakawa-C grid configuration, we establish variable-limiting integral formulations for advection terms, horizontal diffusion terms, and vertical diffusion terms respectively. Stability analysis of the convection equation confirms the robustness of the variable-limiting integral advection scheme. Finally, the variable-limit integral scheme for solving the thermohaline equations is implemented in the Princeton Ocean Model (POM) to demonstrate

its superiority. The stationary isothermal strait simulation demonstrates that compared to the original POM solver, the new variable-limiting integral scheme exhibits smaller temperature-salinity simulation errors. Topographic sensitivity test experi-





ments reveal that as sill steepness increases, the simulation discrepancies between the original POM model (O-POM) and the variable-limiting integral modified POM model (I-POM) progressively widen. Notably, the variable-limiting integral method successfully captures the overflow phenomenon of seawater spilling over sills. In steady-flow experiments of sills with ne-
glected Coriolis force, the implementation of the variable-limiting integral method significantly enhances computational stability in the original POM model and effectively simulates steady-state flows that better align with physical principles. These findings collectively highlight the potential of the variable-limiting integral method in advancing numerical ocean modeling. As a novel stabilized scheme for thermohaline equation solving, the variable-limit integration method developed in this study, while validated in the POM framework, features a universal architecture that is extensible in principle to other ocean numerical
modeling systems.

The remainder of this work is structured as follows. Section 2 outlines the numerical framework for thermohaline transport within the Princeton Ocean Model (O-POM) and provides foundational principles of the variable-limit integral method. Section 3 subsequently details the derivation of characteristic-preserving variable-limit integration schemes for coupled temperature-salinity dynamics, complemented by rigorous von Neumann stability analysis. Section 4 presents the algorithmic implemen-
tation within POM's architecture, including benchmark validation through strategically designed bathymetric test cases and comparative analysis of simulation fidelity. The article concludes with Section 5, which synthesizes key findings, discusses broader implications for ocean model development, and identifies promising directions for future methodological extensions.

## 2  Basic knowledge

In this section, the form of thermohaline equation in ocean motion control equations, the numerical solution of thermohaline
equation in O-POM ocean model and the variable limit integral are introduced.

### 2.1  Thermohaline equation

The potential temperature equation in Reynolds mean conservation form in $\sigma$ coordinates is as follows

$$\frac{\partial DT}{\partial T} + Adv(T) - Dif(T) = \frac{1}{D}\frac{\partial}{\partial \sigma}\left(K_H \frac{\partial T}{\partial T}\right) - \frac{\partial R}{\partial \sigma} \tag{1}$$

where advection term $Adv(T)$ and horizontal diffusion term $Dif(T)$ represent

$$Adv(T) = \frac{\partial(DUT)}{\partial x} + \frac{\partial(DVT)}{\partial y} + \frac{\partial(\omega T)}{\partial \sigma} \tag{2}$$

$$Dif(T) = \frac{\partial}{\partial x}\left(HA_H \frac{\partial T}{\partial x}\right) + \frac{\partial}{\partial y}\left(HA_H \frac{\partial T}{\partial y}\right) \tag{3}$$

$T$ stands for temperature, $D$ stands for stands water height, $H$ stands for water depth, $U, V, \omega$ respectively represent the flow rate in three directions, $A_H$, $K_H$ respectively represent the thermal diffusivity in horizontal and vertical directions. Similarly, the





salinity equation in Reynolds mean conservation form in $\sigma$ coordinates is

$$\frac{\partial DS}{\partial S} + Adv(S) - Dif(S) = \frac{1}{D}\frac{\partial}{\partial \sigma}\left(K_H \frac{\partial S}{\partial S}\right) \tag{4}$$

Advection terms $\mathrm{Adv}(S)$ and horizontal diffusion terms $\mathrm{Dif}(S)$ are consistent with those in the potential temperature equation.

The structural isomorphism between the potential temperature and salinity equations within the thermohaline system permits unified analytical treatment. This study consequently focuses on the temperature equation's semi-implicit discretization

framework in O-POM, with detailed exposition of the variable-limit integral method's high-order discretization methodology. The salinity equation's numerical implementation inherits identical operator structure and weighting strategies, achieving full methodological parity through direct substitution of tracer variables. Such symmetry ensures all derived stability criteria and conservation properties extend equivalently to salinity dynamics without loss of generality.

## 2.2    Solution of temperature and salt equation in POM model

The vertical coordinates of POM ocean model are Arakawa-C grid (C grid for short), which adopts $\sigma$ coordinates along with the terrain and sets the horizontal grid to an interleaving form. The node positions of each variable in POM ocean model C

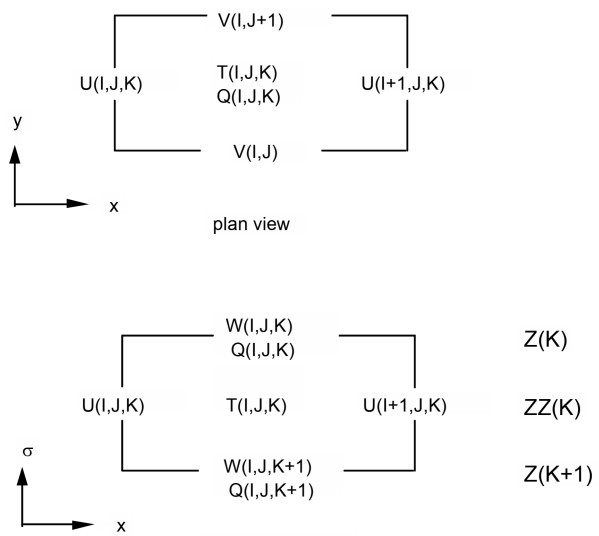

**Figure 1.** POM Ocean Mode 3D Mesh,$Q$ stands for $K_M, K_H$,$T$ stands for $T, S, \rho$ (Figure from POM brochure)

grid are given in figure 1. The potential temperature equation (1) in the Princeton Ocean Model (POM) is numerically solved through a two-stage operator splitting procedure. The first stage implements an explicit finite difference formulation for the horizontal advection and diffusion terms, expressed as:

$$\frac{D^{n+1}\widetilde{T} - D^{n-1}T^{n-1}}{2\Delta t} = -Adv(T^n) + Dif(T^{n-1}) \tag{5}$$





The intermediate variable $\widetilde{T}$ can be calculated directly using the scheme (5), where $\mathrm{Adv}(T^n)$ and $\mathrm{Dif}(T^{n-1})$ use central difference. In POM mode, the process of calculating the intermediate variable $\widetilde{T}$ using format (5) is written as a subroutine advt. The second step is the implicit difference scheme of the vertical diffusion term, as follows

$$\frac{D^{n+1}T^{n+1} - D^{n+1}\widetilde{T}}{2\Delta t} = \frac{1}{D^{n+1}}\frac{\partial}{\partial \sigma}\left(K_H \frac{\partial T^{n+1}}{\partial T}\right) - \frac{\partial R}{\partial \sigma} \tag{6}$$

The format (21) requires an implicit solution for temperature $T^{n+1}$, where the first term on the right uses the central difference scheme. The format (21) for implicitly solving temperature $T^{n+1}$ in the POM ocean model is written as a subroutine proft.

The POM ocean model uses a weak filter (Asselin, 1972) to filter out the computational modes brought by the time-splitting algorithm, and the solution of the thermohaline equation is filtered as follows at each time step

$$T_s = T + \frac{\alpha}{2}(T^{n+1} - 2T^n + T^{n-1}) \tag{7}$$

Where $T_s$ is the filter solution, $\alpha = 0.05$ is usually selected. After the filtering operation (7), $T_s$ will be passed to $T^{n-1}$ and $T^{n+1}$ will be passed to $T^n$.

## 2.3 Variable limit integral method

The variable-limit integral method represents a novel numerical methodology for partial differential equation solutions, recently emerging in computational mathematics. Its defining mechanism employs adaptive integration with variable bounds
(dynamically determined integration limits) within grid node vicinities to formally eliminate spatial derivatives. A canonical formulation of this approach can be expressed through the following operator framework:

$$\int\limits_{xx} f(x) \xlongequal{def} \int\limits_{x_i}^{x_i+\varepsilon_2} dx_b \int\limits_{x_i-\varepsilon_1}^{x_i} dx_a \int\limits_{x_a}^{x_b} f(x)dx. \tag{8}$$

Where $x_i$ is the grid node, variables $x_a$ and $x_b$ are the lower limit and upper limit of the integration respectively, they can be changed, in general, the integration limit parameters $\varepsilon_1$ and $\varepsilon_2$ are required to be greater than zero and the same order as
the grid scale $\Delta x$. For the convenience of the following, we first give the calculation results of some common polynomials' variable-limited integrals. By the definition of variable-limited integrals (8), $a_k$ and $b_k$ can be easily calculated as follows

$$a_k \xlongequal{def} \int\limits_{xx} \frac{(x-x_i)^k}{k!} = \frac{\varepsilon_1 \varepsilon_2^{k+2} + \varepsilon_2(-\varepsilon_1)^{k+2}}{(k+2)!} \tag{9}$$

and

$$b_k \xlongequal{def} \int\limits_{xx} \frac{|x-x_i|^k}{k!} = \frac{\varepsilon_1 \varepsilon_2^{k+2} + \varepsilon_2 \varepsilon_1^{k+2}}{(k+2)!}, \tag{10}$$

Note $\widetilde{a_k} = \frac{a_k}{a_0}, \widetilde{b_k} = \frac{b_k}{a_0}$ where the script $k$ is a non-negative integer. Let's give the lemma





**Lemma 1.** *suppose $f(x) \in C^{K+1}[x_l, x_r]$, then*

$$\int_{xx} f(x) = \sum_{k=0}^{K} a_k f_i^{(k)} + R$$

*and*

$$|R| \leq \max_{x \in [x_l, x_r]} \left| f^{(K+1)}(x) \right| b_{K+1}$$

*where $f_i^{(k)} = \frac{d^k f}{dx^k}|_{x=x_i}$, $k \in \mathbb{N}$.*

The lemma gives a method to approximate the variable limit integral of $\int_{xx}$ based on the Taylor expansion of the integrand.

## 3   Design of variable limit integral scheme for thermohaline equation

This subsection systematically develops variable-limit integral numerical schemes for the temperature equation's horizontal
advection, horizontal diffusion, and vertical diffusion terms within the Arakawa-C grid framework, complemented by rigorous
von Neumann stability analyses.

### 3.1   Variable limit integral scheme design for advection term $\mathbf{Adv}(T)$

Taking $\frac{\partial(DUT)}{\partial x}$ as an example, the discretization process of advection term $\mathrm{Adv}(T)$ by variable limit integral method is
introduced. Since $D$ and $T$ are on the same grid and differ by half a step from $U$, we can substitute $e$ for $DT$ and $u$ for $U$, and
consider the following equation

$$p = \frac{\partial(ue)}{\partial x} \tag{11}$$

The grid of $p$ and $e$ is the same, and the grid corresponding to $u$ and $e$ is shown in Figure 2.

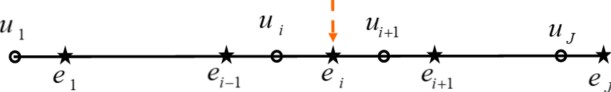

**Figure 2.** C grid diagram of variables $u, e$

Use the auxiliary function $e(x)$ at node $x_{i-1}, x_i, x_{i+1}$; where the function values are $e_{i-1}, e_i, e_{i+1}$, a three-point Lagrange
interpolation function can be constructed near node $x_i$ to approximate the auxiliary function $e(x)$.

$$e(x) = \sum_{k=i-1}^{k=i+1} e_k L_k(x) + \Delta x^3$$





Where $L_k(x)$ is the Lagrange interpolation basis function at node $x_k$. Similarly, since $p$ and $e$ mesh are the same, a similar three-point Lagrange interpolation function can be constructed to approximate the auxiliary function $p(x)$.

$$p(x) = \sum_{k=i-1}^{k=i+1} p_k L_k(x) + \Delta x^3$$

For function $u$, it can be based on node $x_{i-\frac{1}{2}}$, Function value $u_i, u_{i+1}$ constructs a two-point Lagrange interpolation function near node $x_i$ to approximate

$$u(x) = -u_i \frac{1}{\Delta x}(x - x_i - \Delta x) + u_{i+1}\frac{1}{\Delta x}(x - x_i + \Delta x) + \Delta x^2.$$

Next, the variable limit integral of $\int_{xx}$ (as defined by equation (8)) on both sides of equation (11), $p, u, e$ is replaced by the interpolation function, the integral is calculated and sorted, and the final variable limit integral format is as follows


$$(\frac{\widetilde{a_2}}{\Delta x^2} - \frac{\widetilde{a_1}}{2\Delta x})p_{i-1} + (1 - \frac{2\widetilde{a_2}}{\Delta x^2})p_i + (\frac{\widetilde{a_2}}{\Delta x^2} + \frac{\widetilde{a_1}}{2\Delta x})p_{i+1} = \delta(u_i, e_i) + \Delta x^2 \tag{12}$$

where

$$\begin{aligned}
\delta(u_i, e_i) = &\frac{1}{4\Delta x}(u_i + u_{i+1})(e_{i+1} - e_{i-1}) + \frac{1}{\Delta x}e_i(u_{i+1} - u_i) \\
&+ \frac{\widetilde{a_1}}{2\Delta x^2}(u_{i+1} - u_i)(e_{i+1} - e_{i-1}) + \frac{\widetilde{a_1}}{\Delta x^2}((e_{i+1} - e_i)u_{i+1} - (e_i - e_{i-1})u_i) \\
&+ \frac{3\widetilde{a_2}}{\Delta x^3}(u_{i+1} - u_i)(e_{i+1} - 2e_i + e_{i-1})
\end{aligned}$$

The following uses 1 to give a fourth-order variable-limit integral scheme for equation (11). The variable limit integral is applied to both sides of equation (11), 1 is applied and $\varepsilon_1 = \varepsilon_2 = \varepsilon$ is obtained


$$p_i + \frac{\varepsilon^2}{12}p_i^{(2)} = (ue)_i^{(1)} + \frac{\varepsilon^2}{12}(ue)_i^{(3)} + O(\varepsilon^4) \tag{13}$$

Easy to know by Taylor's formula

$$p_i^{(2)} = \frac{p_{i+1} - 2p_i + p_{i-1}}{\Delta x^2} + O(\Delta x^2)$$

$$(ue)_i^{(1)} + \frac{\Delta x^2}{6}(ue)_i^{(3)} = \delta^*(u_i, e_i) + O(\Delta x^4)$$

where

$$\delta^*(u_i, e_i) = \frac{(u_{i-1} - 5u_i + 15u_{i+1} + 5u_{i+2})e_{i+1} - (5u_{i-1} + 15u_i - 5u_{i+1} + u_{i+2})e_{i-1}}{32\Delta x}$$

Let $\varepsilon = \sqrt{2}\Delta x$, As can be seen from the above two formulas, equation(13) can be reduced to

$$\frac{1}{6}(p_{i+1} + 4p_i + p_{i-1}) = \delta^*(u_i, e_i) + O(\Delta x^4) \tag{14}$$

**Remark 1.** *Format (12) and format (14) need to be combined with boundary conditions to solve and calculate the corresponding $p_i$.*





## 3.2  Variable limit integral scheme design for horizontal diffusion term $\mathrm{Dif}(T)$

The diffusion term $\frac{\partial}{\partial x}\left(HA_H\frac{\partial T}{\partial x}\right)$ in the $x$ direction is taken as an example to introduce the discretization process of the variable limit integral method of the horizontal diffusion term $\mathrm{Dif}(T)$. The diffusion term in the $y$ direction is similar and will not be repeated here. Easy to know

$$\frac{\partial}{\partial x}\left(HA_H\frac{\partial T}{\partial x}\right)=\frac{\partial(HA_H)}{\partial x}\frac{\partial T}{\partial x}+(HA_H)\frac{\partial^2 T}{\partial x}$$

Since $H$ $A_H$and $T$ are on the same grid, the key to discretization of diffusion term $\mathrm{Dif}(T)$ by variable-limited integral method is how to discretize the first and second derivatives by variable-limited integral method. So let's start with the first derivative, for the equation below

$$p=\frac{\partial(e)}{\partial x} \tag{15}$$

Take both sides of the variable limit integral and apply 1 and take $\varepsilon_1=\varepsilon_2=\varepsilon$ to obtain

$$p_i+\frac{\varepsilon^2}{12}p_i^{(2)}=(e)_i^{(1)}+\frac{\varepsilon^2}{12}(e)_i^{(3)}+O(\varepsilon^4) \tag{16}$$

Easy to know by Taylor's formula

$$p_i^{(2)}=\frac{p_{i+1}-2p_i+p_{i-1}}{\Delta x^2}+O(\Delta x^2)$$

$$(e)_i^{(1)}+\frac{\Delta x^2}{6}(e)_i^{(3)}=\delta_1(e_i)+O(\Delta x^4)$$

where

$$\delta_1(e_i)=\frac{e_{i+1}-e_{i-1}}{2\Delta x}$$

Let$\varepsilon=\sqrt{2}\Delta x$, and from the above two formulas, formula (16) can be reduced to

$$\frac{1}{6}(p_{i+1}+4p_i+p_{i-1})=\delta_1(e_i)+O(\Delta x^4) \tag{17}$$

Similarly, for the second derivative, consider the following equation

$$q=\frac{\partial^2(e)}{\partial x^2} \tag{18}$$

The variable limit integral is applied to both sides of equation (18), and 1 is applied and $\varepsilon_1=\varepsilon_2=\varepsilon$ is obtained

$$p_i+\frac{\varepsilon^2}{12}p_i^{(2)}=(e)_i^{(2)}+\frac{\varepsilon^2}{12}(e)_i^{(4)}+O(\varepsilon^4) \tag{19}$$

Easy to know by Taylor's formula

$$p_i^{(2)}=\frac{p_{i+1}-2p_i+p_{i-1}}{\Delta x^2}+O(\Delta x^2)$$

$$(e)_i^{(2)}+\frac{\Delta x^2}{12}(e)_i^{(4)}=\delta_2(e_i)+O(\Delta x^4)$$





where

$$\delta_2(e_i) = \frac{e_{i+1} + 2e_i + e_{i-1}}{\Delta x^2}$$

Let $\varepsilon = \Delta x$, and from the above two formulas, formula (19) can be reduced to

$$\frac{1}{12}(p_{i+1} + 10p_i + p_{i-1}) = \delta_2(e_i) + O(\Delta x^4) \tag{20}$$

**Remark 2.** *The first and second order discrete derivatives $P_i$ and $q_i$ can be calculated from equations (17) and (20) and the corresponding boundary conditions.*

### 3.3 Variable limit integral scheme design and implicit discretization of vertical diffusion term

The vertical diffusion term $\frac{\partial}{\partial \sigma}\left(K_H \frac{\partial T^{n+1}}{\partial T}\right)$ in POM ocean model adopts central difference scheme. The variable limit integral scheme design of vertical diffusion term is introduced below. Consider the following time-implicit difference scheme for the

vertical diffusion term in the POM model

$$\frac{D^{n+1}T^{n+1} - D^{n+1}\widetilde{T}}{2\Delta t} = \frac{1}{D^{n+1}}\frac{\partial}{\partial \sigma}\left(K_H \frac{\partial T^{n+1}}{\partial \sigma}\right) - \frac{\partial R}{\partial \sigma} \tag{21}$$

Notice that $K_H$ and $T$ are not on the same grid in the direction $\sigma$ and are half a step apart. To facilitate derivation, the above formula can be rewritten as

$$Q = \frac{1}{D^{n+1}}\frac{\partial}{\partial \sigma}\left(K_H \frac{\partial T^{n+1}}{\partial \sigma}\right) \tag{22}$$

For auxiliary function $Q$ and temperature $T$ at node $z_{k-1}, z_k, z_{k+1}$ Construct a three-point interpolation function, for $K_H$, two-point interpolation function is constructed at node $z_{k-\frac{1}{2}}, z_{k+\frac{1}{2}}$, and the variable limiting integral of $\int\limits_{\sigma\sigma}$ (defined as equation (8), where the integral variable is $\sigma$) is used on both sides of equation (22). $Q, T, K_H$ is replaced by the corresponding interpolation function, the integral is calculated and sorted, and the following final variable limit integral format is obtained

$$\Re(Q_k) = \frac{1}{D^{n+1}}\left(\alpha_0 T_{k+1}^{n+1} + \alpha_1 T_k^{n+1} + \alpha_2 T_{k-1}^{n+1}\right) \tag{23}$$

where the operator $\Re$ on the left is

$$\begin{aligned}
\Re(Q_k) =& (\frac{\widetilde{a_2}}{\Delta x^2} - \frac{\widetilde{a_1}}{2\Delta x})Q_{k+1} + (1 - \frac{2\widetilde{a_2}}{\Delta x^2})Q_k + (\frac{\widetilde{a_2}}{\Delta x^2} + \frac{\widetilde{a_1}}{2\Delta x})Q_{k-1} \\
&\doteq \beta_0 Q_{k+1} + \beta_1 Q_k + \beta_2 Q_{k-1}
\end{aligned} \tag{24}$$

The coefficient of the right end of equation (23) $\alpha_0, \alpha_1, \alpha_2$ in turn is

$$\alpha_0 = \frac{1}{d_z^2}K_H(k-\frac{1}{2}) - \frac{2\widetilde{a_1}}{d_z^3}\left(K_H(k-\frac{1}{2}) - K_H(k+\frac{1}{2})\right)$$

$$\alpha_1 = -\frac{1}{d_z^2}\left(K_H(k-\frac{1}{2}) + K_H(k+\frac{1}{2})\right) - \frac{4\widetilde{a_1}}{d_z^3}\left(K_H(k-\frac{1}{2}) - K_H(k+\frac{1}{2})\right)$$

$$\alpha_2 = \frac{1}{d_z^2}K_H(k+\frac{1}{2}) - \frac{2\widetilde{a_1}}{d_z^3}\left(K_H(k-\frac{1}{2}) - K_H(k+\frac{1}{2})\right)$$



let $\frac{D^{n+1}T^{n+1} - D^{n+1}\widetilde{T}}{2\Delta t} + \frac{\partial R}{\partial \sigma}$ replace $Q$, and then the terms containing $T^{n+1}$ are grouped together, the formula (23) This can be

converted to the following final variable-limit integral implicit fully discrete scheme

$$\gamma_{k+1}T_{k+1}^{n+1} + \gamma_k T_k^{n+1} + \gamma_{k-1}T_{k-1}^{n+1} = P_k \tag{25}$$

where

$$\gamma_{k+1} = \frac{D^{n+1}}{2\Delta t}\beta_0 - \frac{\alpha_0}{D^{n+1}}$$

$$\gamma_k = \frac{D^{n+1}}{2\Delta t}\beta_1 - \frac{\alpha_1}{D^{n+1}}$$

$$\gamma_{k-1} = \frac{D^{n+1}}{2\Delta t}\beta_2 - \frac{\alpha_2}{D^{n+1}}$$

$$P_k = \frac{D^{n+1}}{2\Delta t}\Re(\widetilde{T}) - \Re(\frac{\partial R}{\partial \sigma})$$

**Remark 3.** *The temperature Tn+1 at all discrete points can be obtained by combining the two boundary conditions of tem-*

*perature at the sea surface and bottom of equation 25. The setting and use of boundary conditions are the same as POM ocean*

*model.*

### 3.4 Stability analysis of variable limit integral scheme

To directly compare the performance of variable-limit integration schemes, this study adopts the same temporal discretization as the POM ocean model, employing a three-time-level leap-frog scheme. For spatial discretization, variable-limit integration

schemes are designed for horizontal advection terms, horizontal diffusion terms, and vertical diffusion terms. Specifically, the vertical diffusion term combines a variable-limit integration scheme with a finite difference method to form an implicit computational scheme. While implicit schemes generally exhibit superior stability, horizontal advection terms often significantly influence the stability of the numerical scheme for the temperature and salinity equations. Therefore, this section focuses on analyzing the stability of the variable-limit integration scheme for horizontal advection terms under the three-time-level leap-frog

temporal discretization.

Consider the one-dimensional constant-coefficient convection equation

$$\frac{\partial e}{\partial t} = u\frac{\partial e}{\partial x} \tag{26}$$

where $u$ is constant.

Building upon the three-time-level temporal discretization scheme of the Princeton Ocean Model (5) and the variable-limit

integral formulation for horizontal transport terms (12), we derive the fully discrete scheme for Equation (26)

$$\mathcal{A}\frac{e_i^{n+1} - e_i^{n-1}}{2\Delta t} = \delta(u, e_i) \tag{27}$$

where the discrete operators are defined as

$$\mathcal{A}p_i = \left(\frac{\widetilde{a_2}}{\Delta x^2} - \frac{\widetilde{a_1}}{2\Delta x}\right)p_{i-1} + \left(1 - \frac{2\widetilde{a_2}}{\Delta x^2}\right)p_i + \left(\frac{\widetilde{a_2}}{\Delta x^2} + \frac{\widetilde{a_1}}{2\Delta x}\right)p_{i+1}$$

$$\delta(a, e_i) = \frac{u}{2\Delta x}(e_{i+1} - e_{i-1}) + \frac{\widetilde{a_1}u}{\Delta x^2}(e_{i+1} - 2e_i - e_{i-1})$$




Notably, the fourth-order variable-limit integral scheme (14) is inherently contained within the discrete formulation (27) for
constant-coefficient convection problems. Consequently, the subsequent stability analysis encompasses both second-order (12)
and fourth-order (14) variable-limit integral schemes for horizontal transport terms.

Following conventional practice in variable-limit integral formulations (9), we normalize the integral limit parameter $\varepsilon$ to
match the grid scale $\Delta x$ order, establishing

$$\rho_1 = \frac{\widetilde{a_1}}{\Delta x}, \quad \rho_2 = \frac{\widetilde{a_2}}{\Delta x^2}$$

Introducing the auxiliary variable $w_i^n = e_i^{n-1}$ and applying Fourier analysis with the ansatz $e_i^n = e^n \exp(i\alpha x_i)$, $w_i^n = w^n \exp(i\alpha x_i)$, substitution into Equation (27) yields after common factor elimination

$$e^{n+1} = 2ur\frac{B}{A}e^n + w^n$$
$$w^{n+1} = e^n \tag{28}$$

where $r \equiv \Delta t/\Delta x$ denotes the grid ratio, and coefficients $A$, $B$ are given by

$$A = (\rho_2 - \frac{1}{2}\rho_1)\exp(-i\alpha\Delta x) + (1 - 2\rho_2) + (\rho_2 + \frac{1}{2}\rho_1)\exp(i\alpha\Delta x)$$
$$B = (\rho_1 - \frac{1}{2})\exp(-i\alpha\Delta x) - 2\rho_1 + (\rho_1 + \frac{1}{2})\exp(i\alpha\Delta x)$$

The corresponding amplification matrix becomes

$$G = \begin{pmatrix} 2ur\frac{B}{A} & 1 \\ 1 & 0 \end{pmatrix}$$

Letting $d \equiv ur\frac{B}{A}$, the characteristic equation of matrix $G$ is

$$\lambda^2 - 2d\lambda - 1 = 0$$

yielding eigenvalues

$$\lambda = d \pm \sqrt{d^2 + 1}$$

Notably, $d$ is generally complex-valued. Through Euler's formula with $\theta \equiv \alpha\Delta x$, we simplify $\frac{B}{A}$ as

$$\frac{B}{A} = \frac{2\rho_1(\cos\theta - 1) + i\sin\theta}{2\rho_2(\cos\theta - 1) + \rho_1 i\sin\theta + 1}$$

Direct verification confirms that $|\lambda| \leq 1$ holds when $\rho_1 = 0$, $\rho_2 = \frac{1}{6}$, and $r \leq \frac{1}{u\sqrt{3}}$, thereby establishing numerical stability.

## 4 Numerical Experiments

This section presents the computational implementation strategy for integrating the variable-limit integral scheme into the host
ocean model's codebase, accompanied by verification procedures. Validation employs two benchmark test cases: an idealized



straight-channel configuration and a zero-Coriolis scenario, designed to rigorously assess the scheme's numerical reliability and long-term stability. The analysis concludes with a systematic evaluation of simulation accuracy, comparing the proposed scheme against conventional model formulations under diverse steep bathymetric conditions (slope steepness ratios exceeding 0.15), thereby demonstrating the method's enhanced topographic adaptability.

Leveraging the distinct mathematical characteristics of the thermohaline equation's horizontal advection, horizontal diffusion, and vertical diffusion operators, this study develops specialized variable-limit integral formulations under the C-grid discretization framework. The implementation modifies the host ocean model's Fortran architecture through three critical enhancements.

The calculation result of I-POM mentioned in the following numerical experiment refers to replacing the calculation part of the temperature and salt equation of the original POM model (that is, the subroutine advt1 or advt2 and the subroutine proft) with the designed variable limit integral (subroutine advtle and proftle), and the other parts continue to follow the POM model. In the following numerical experiments, we use O-POM to represent the results of the original POM model and I-POM to represent the results of the POM model improved by the variable limit integration method.

## 4.1 The Setting of Numerical Experiment

### 4.1.1 Terrain Settings

In the following numerical example, only two types of topography, the strait and the sea sill, are used, as shown in figure 3. The experimental domain is designed as an idealized strait geometry with zonal (east-west) and meridional (north-south) extents of 520 km and 400 km, respectively; bathymetric characterization reveals a maximum water depth of 4,500 m in the central basin. Boundary conditions are prescribed with dynamically constrained no-flux walls along latitudinal boundaries (north-south orientation) and free-slip open boundaries permitting cross-strait exchange in longitudinal directions (east-west orientation).

Sill topography is based on the setting of the channel, the flat seafloor is replaced with a gradually raised sill. The east-west length is still 520km, the north-south width is 400km, and the depth is a gradually changing sill of 4500m. The topography (depth) function h(x; y) as follows,

$$h(x,y) = 4500 \left(1 - A / \cosh(\alpha(x - x_m))\right) \tag{29}$$

Where $\alpha = \frac{40}{512000}$, $x_m = 260000$, $A$ is the steepness of the terrain parameter value is $0 \le A \le 1$, $A$, the larger the A indicates the steeper the terrain. Similarly, the northern and southern boundaries of the area are closed and impenetrable, and the eastern and western boundaries are open to allow water to flow in and out freely.





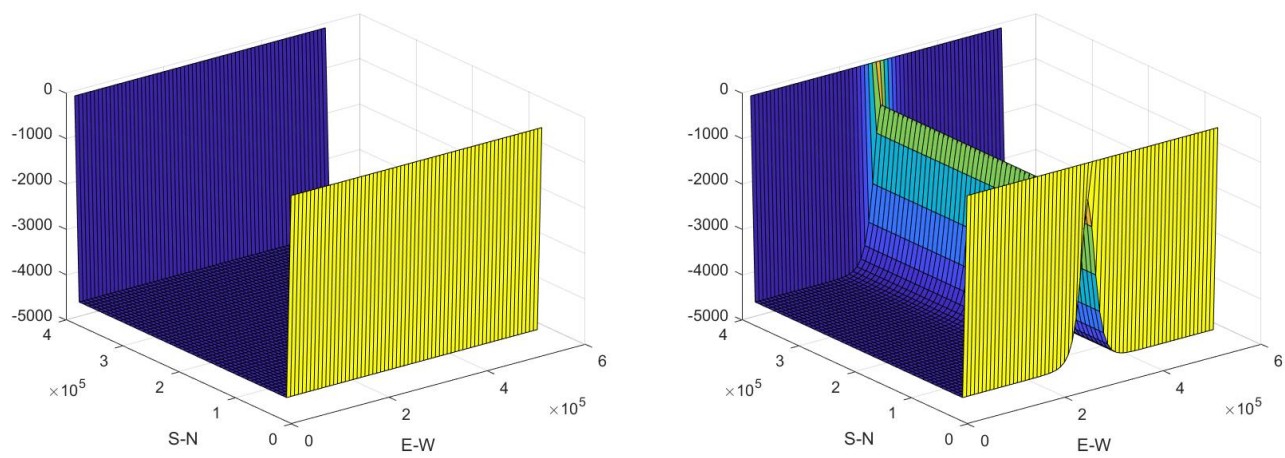

**Figure 3.** Shematic diagram of the channel and sea sill

### 4.1.2 Temperature and salt flow rate and grid setting

The temperature and salinity of the initial seawater in the region and the seawater flowing in from the western boundary are set in two ways: 1. The temperature is set to a constant value of 20°C, and the salinity is set to a constant value of 35psu (channel zero start-up example); 2. Temperature and salinity change with depth (other examples). Specifically, sea water temperature is only related to depth $z$ and decreases with increasing depth, and the temperature distribution function $t_0(z)$ is as follows

$$t_0(z) = 5 + 25\exp\left(z/1000\right) \tag{30}$$

Seawater salinity is also only related to seawater depth $z$, set to increase with depth, and the salinity distribution function $s_0(z)$ is as follows

$$s_0(z) = 35 - \exp\left(z/1000\right) \tag{31}$$

With regard to the flow rate, in the example, it is either set to no seawater inflow (channel zero start-up example), or set to a constant flow at the western boundary (other examples), flowing from west to east into the channel (sill) area at a speed of

$0.2m/s$. As for the grid Settings, unless otherwise specified, the horizontal grid resolution in the numerical example in this paper is 8km, that is, $dx = dy = 8000$(m); The vertical coordinates are $\sigma$ with the terrain, and the vertical directions are divided into 20 parts.





## 4.2 numerical experiment

### 4.2.1 Channel zero start-up example

This part of the calculation uses the channel topography, the temperature is set at a constant $20°C$, the salinity is set at a constant 35 psu, no seawater flows in or out of the boundary, and the water in the entire channel remains static. From this, the mode is activated and, in theory, the temperature and salinity are always constant at any moment, and the flow rate is always zero.

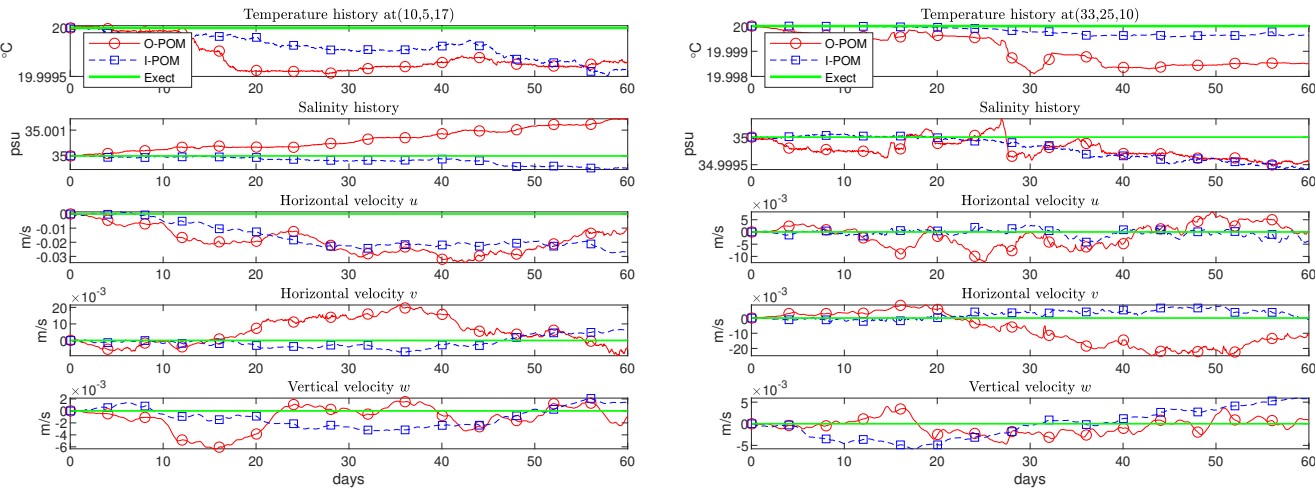

**Figure 4.** Temperature, salinity and velocity timelines of I-POM and O-POM at locations (10,5,17) and (33,25,10) under channel test

Figure 4 compares the simulated temperature, salinity, and velocity fields generated by the Princeton Ocean Model (O-POM)
and the variable-limit integral method (I-POM) in an open-channel configuration. The left panels present temporal evolution patterns at coordinate position (10, 5, 17), while the right panels display corresponding temporal variations at coordinate (33, 25, 10). Here, the indices (i, j, k) denote the spatial position ($x = i\Delta x, y = j\Delta y, \sigma = k$) within the sigma-coordinate system, where $\sigma$ represents the dimensionless vertical coordinate.

Figure 5 quantifies model discrepancies through error norm analysis, with left panels illustrating $L^2$-error norms and right
panels depicting $L^\infty$-error norms for temperature, salinity, and velocity components. These error metrics systematically compare the numerical differences between O-POM and I-POM simulations.

According to the model configuration, constant temperature ($20°C$) and salinity (35 psu) conditions were maintained, with zero velocity prescribed in all three spatial directions. As illustrated in Figure 4 and Figure 5, both the Princeton Ocean Model (O-POM) and the variable-limit integral IMVL method (I-POM) successfully reproduced the temperature, salinity, and flow
fields under this static water scenario.





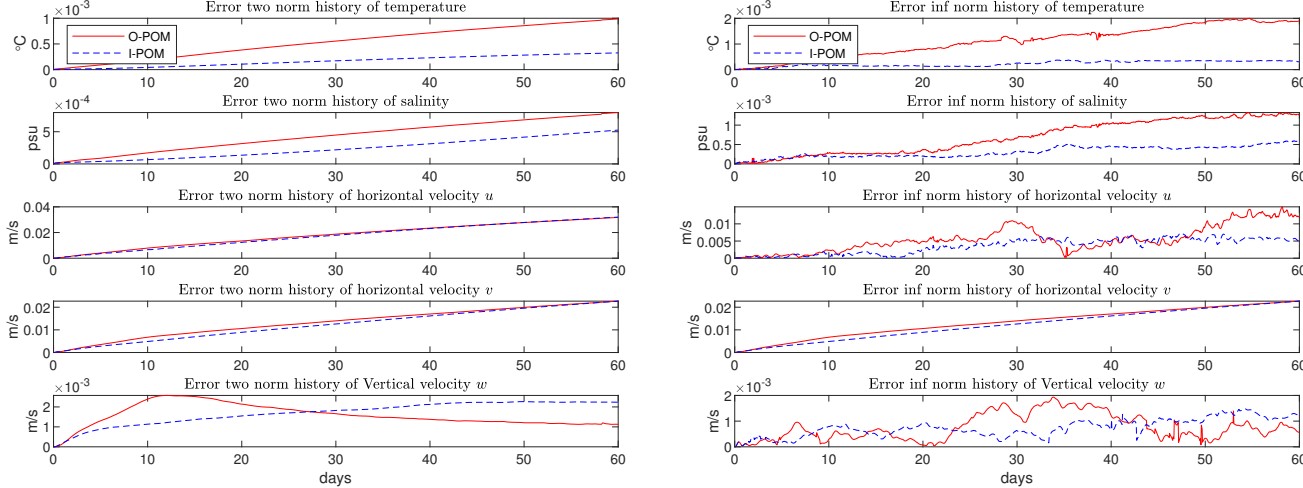

**Figure 5.** The time history graph of temperature salinity velocity $L^2$-error (left) and $L^\infty$-error (right) calculated by I-POM and O-POM under channel test

A detailed comparison reveals that the I-POM approach achieved higher precision in simulating temperature and salinity distributions compared to the O-POM (Figure 4). Error analysis in Figure 5 further demonstrates that the I-POM's global error accumulated at a slower rate over time. By day 60, the I-POM's error magnitude was reduced by approximately 50% relative to the O-POM baseline. These results indicate that the introduction of variable-limit integration significantly enhances the ac-
curacy of thermohaline simulations. For hydrodynamic components, however, both models exhibited comparable performance in reproducing horizontal velocities ($u$, $v$) and vertical velocity ($w$), with minimal discrepancies observed between the two methodologies.

### 4.2.2    Runtime Comparison

The computational performance of the Integral Method with I-POM and the O-POM was systematically compared under differ-
ent spatial resolutions. Table 1 presents the computational time distribution for three progressively refined grid configurations($I_m \times J_m \times K_m$): $(33 \times 25 \times 11), (65 \times 49 \times 21), and (129 \times 99 \times 41)$, where $I_m$ (zonal), $J_m$ (meridional), and $K_m$(vertical) denote the grid node counts in respective coordinate directions. The timing analysis distinguishes between total computation duration (All), temperature-salinity component (STp), and the IMVL module, with all times reported in seconds (s).





**Table 1.** Total calculation time of I-POM and O-POM under zero Coriolis force test with different spatial resolutions $(im, jm, kb)$ All and temperature and salinity calculation time STp (unit: second)

|  | (33×25×11) | | (65×49×21) | | (129×99×41) | |
|---|---|---|---|---|---|---|
|  | All | STp | All | STp | All | STp |
| POM | 23.050 | 2.882 | 145.958 | 21.408 | 1157.556 | 186.579 |
| I-POM | 27.979 | 8.478 | 187.322 | 63.772 | 1470.236 | 531.552 |
| Rate of increase | 21.38% | 194.17% | 28.34% | 197.89% | 27.01% | 184.89% |

Key findings reveal that the I-POM implementation requires approximately three times longer computation duration than
conventional O-POM approaches when solving the temperature-salinity equations. Notably, substituting the O-POM's native temperature-salinity solver with the IMVL scheme increases the model's overall computational overhead by approximately 25%. This performance differential remains consistent across all tested grid resolutions, demonstrating the method's scalability characteristics.

## 4.3 The results of I-POM and O-POM numerical models are different in different steep sill topography

This section presents a numerical experiment configured with a sill topography. The temperature and salinity fields are prescribed according to Equations (30) and (31), respectively. A steady westward inflow of 0.2 m/s is imposed along the western boundary to drive the circulation through the sill region. The Coriolis parameter is specified as $cor = 1.0 \times 10^{-4}\,\text{s}^{-1}$. To investigate the discrepancies between variable limit integration (I-POM) and the original Princeton Ocean Model (O-POM) in temperature-salinity simulations under different topographic gradients, the topographic steepness parameter $A$ in the bathymetry function is systematically varied with three distinct values: 0.4, 0.6, and 0.8. This parametric study enables comparative analysis of model performance across gradually intensified topographic slopes.

Figure 6 and Figure 7, The temperature and salinity cross sections of the I-POM (left) and POM (right) at the central axial plane y=196km under the sea sill test are shown, respectively. The operating days from top to bottom are 5 days, 30 days, and 60 days, respectively, where the steepness parameter $A = 0.6$. As can be seen from the figure, there is little difference between the results of I-POM and O-POM on the whole, regardless of temperature or salinity. However, with the increase of simulation time, the difference between the two begins to appear: in the 30-day and 60-day sections, the bottom of the sill, especially the downstream part ($x \geq 300$), is significantly different. In addition, in the figure7, The salinity cross section (left 3 and left 4) is simulated by the I-POM on the 30th and 60th days of the presentation in the above paper. It can be seen that there are multiple isosalinity curves that intersect at the downstream profile along the top of the sill, and the isosalinity curve at the top of the left 4 fluctuates dissimilarly, indicating that the variable limit integral I-POM can simulate the overflow mixing of seawater here. More in line with the actual seawater movement; In the salinity cross section of I-POM 60 days, overflow mixing even affects the surface seawater, while O-POM does not simulate similar results.




**Figure 6.** Temperature cross sections at y=196km for I-POM (left), POM (middle) and the difference between them (right) under sea sills







**Figure 7.** Salinity cross sections at y=196km of I-POM (left), POM (middle) and the difference between them (right) under sea sills







**Figure 8.** Temperature cross sections at y=196km of I-POM (left) and O-POM (right) on day 18 under sea sills with different steepness parameters $A = 0.4$(top),0.6(middle),0.8(bottom)

    Figures 8 and 9 respectively present the simulated temperature and salinity cross-sections obtained using the variable-limit integral method (I-POM, left panels) and the Princeton Ocean Model (O-POM, right panels) under varying sill steepness conditions. The steepness coefficient A progressively increases from 0.4 (top row) to 0.6 (middle row) and 0.8 (bottom row). All cross-sections were extracted from the central axial plane (y = 196 km) of the computational domain at identical simulation times. Figure 10 illustrates vertical velocity profiles along three orthogonal directions: longitudinal (u, left), lateral (v, center), and vertical (w, right). These profiles compare the I-POM and POM model outputs under different sill steepness conditions (A = 0.4, 0.6, 0.8 from top to bottom) at day 18 of the simulation. Consistent with Figures 8-9, all velocity profiles were sampled at the central cross-section (y = 196 km).





**Figure 9.** Salinity cross sections at y=196km of I-POM (left) and O-POM (right) on day 18 under sea sills with different steepness parameters $A = 0.4$(top),0.6(middle) and 0.8(bottom)





**Figure 10.** Velocity $u, v, w$ difference between I-POM and O-POM on day 18 under sea sills with different steepness parameters $A =$ 0.4(top),0.6(middle) and 0.8(bottom)

The figure 8,9 and10 demonstrates that both temperature and salinity fields simulated by the I-POM and O-POM exhibit general consistency under varying steepness coefficients. However, as the steepness coefficient A increases from 0.2 to 0.8, discrepancies between the two methods become progressively more pronounced, particularly in the sill's basal regions ($x \geq$ 300) of the downstream section. Notably, when A=0.8, the variable limit integral method successfully captures critical hydraulic overflow processes induced by abrupt topographic uplift, along with the intense turbulent mixing phenomena occurring post-sill crossing. In contrast, the O-POM fails to reproduce these dynamic features under equivalent parameter conditions.





### 4.3.1 Zero Coriolis force sill test

In this set of experiments, we configure the Coriolis parameter (denoted as two-dimensional variable *cor* in the O-POM model codes) to zero, effectively eliminating Coriolis force effects. The strait topography is maintained with initial temperature and

salinity distributions prescribed by Equations (30) and (31), respectively. A steady westward inflow of 0.2 m/s is imposed at the western boundary. Under this Coriolis-free configuration, symmetry considerations dictate that all thermohaline-current physical quantities should remain uniform along the north-south direction at any horizontal level.

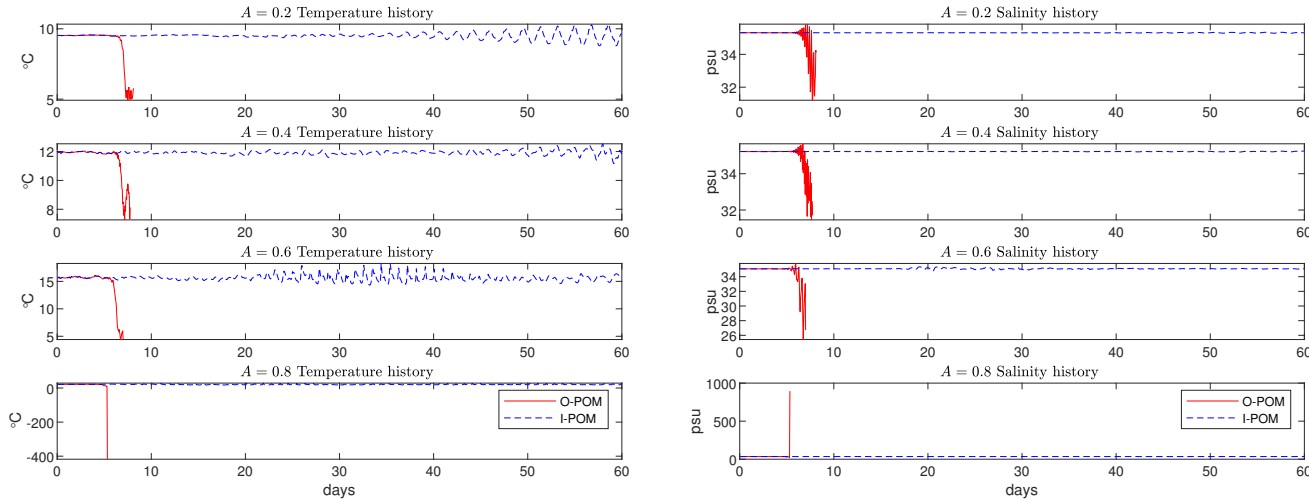

**Figure 11.** Temperature (left) and salinity (right) time graphs of I-POM and O-POM with different steep terrain parameters $A$ under zero Coriolis force sill test

Figure 11 illustrates temporal variations of temperature (left panel) and salinity (right panel) at the strait center point simulated by both O-POM and I-POM approaches, with topographic steepness parameter A decreasing from 0.2 (top) to 0.8

(bottom).

Our initial objective was to evaluate I-POM's performance in thermohaline simulations by verifying preservation of north-south uniformity. However, we uncovered a noteworthy phenomenon: The implementation of variable-limit integration (IMVL) significantly enhances numerical stability compared to the O-POM model under these configurations.

As evidenced in Figure 11, the standard O-POM model exhibits numerical instability across all tested A values when Coriolis

effects are disabled. Temperature and salinity simulations develop pronounced oscillations around days 6-8, followed by rapid divergence that ultimately forces code termination due to excessive horizontal velocity v. In contrast, IMVL demonstrates remarkable stability with only minor oscillations (potentially related to velocity solution mechanisms), maintaining robust computation throughout the simulation period.





To investigate whether the numerical instability observed in the O-POM is terrain-induced, we conducted stability tests by
removing the bottom topography (i.e., reverting to a flat-bottom channel configuration) under zero Coriolis effects and identical
temperature-salinity-current settings. The simulations compared the O-POM and the I-POM.

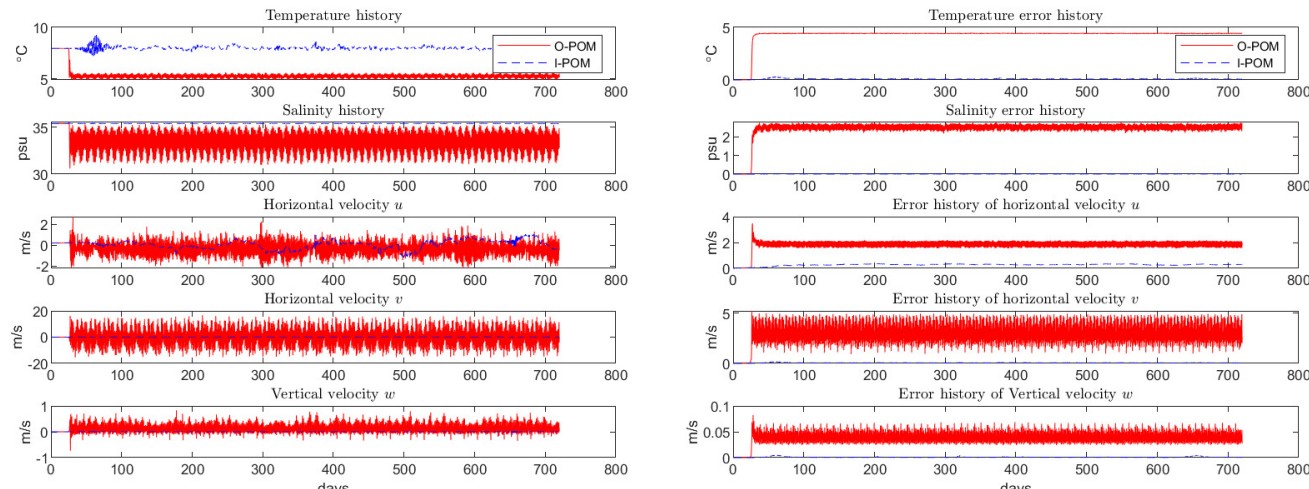

**Figure 12.** Temperature, salinity and velocity time history at locations (33,25,10) (left) and mean absolute errors (right) graph of I-POM and
O-POM under zero Coriolis force sill test when $A = 0$.

Figure 12 presents the temporal variations of mean absolute errors (MAEs) in temperature, salinity, and velocity between
the two methods over the same period. The MAE for temperature, denoted as $E_t(n)$, is computed as

$$E_t(n) = \frac{\sum_{i,j,k} |t_{i,j,k} - t_{i,j,k}^{\text{exact}}|}{I_m \cdot J_m \cdot K_m},$$

where $t_{i,j,k}$ and $t_{i,j,k}^{\text{exact}}$ represent simulated and exact solutions, respectively. Analogous formulas apply to salinity and three-
dimensional velocity errors.

The results reveal that while the O-POM simulation remains operational, it exhibits severe numerical oscillations in temper-
ature, salinity, and velocity around day 25. In contrast, the I-POM maintains stable solutions close to the ground truth, with
only minor oscillations occurring near day 50. Error analysis confirms that O-POM-derived results become unreliable after
approximately 25 days, whereas I-POM maintains consistently low MAEs throughout the 720-day simulation. Figures 11 and
12 collectively demonstrate that the IMVL formulation significantly enhances numerical stability in the temperature-salinity
equations compared to the O-POM framework.

395 **4.3.2 Stability Test under Sill Terrain**

Table 2 presents the stability comparison between POM and IMVL under different Coriolis force parameters (presence/ab-
sence) and topographic conditions (sill topography vs. strait topography). Initial analysis indicates that the POM scheme




exhibits significant numerical instability under Coriolis-free conditions, while topographic effects can partially mitigate such instabilities.

**Table 2.** Stability comparison between O-POM and I-POM under different Coriolis force parameters and topographic conditions

| Condition Type | O-POM | I-POM |
|---|---|---|
| No Coriolis Force & sill | Crash | Normal |
| No Coriolis Force & channel | Deviation in calculation around 25 days, results not reliable | Normal |
| With Coriolis Force & sill | Normal | Normal |
| With Coriolis Force & channel | Normal | Normal |

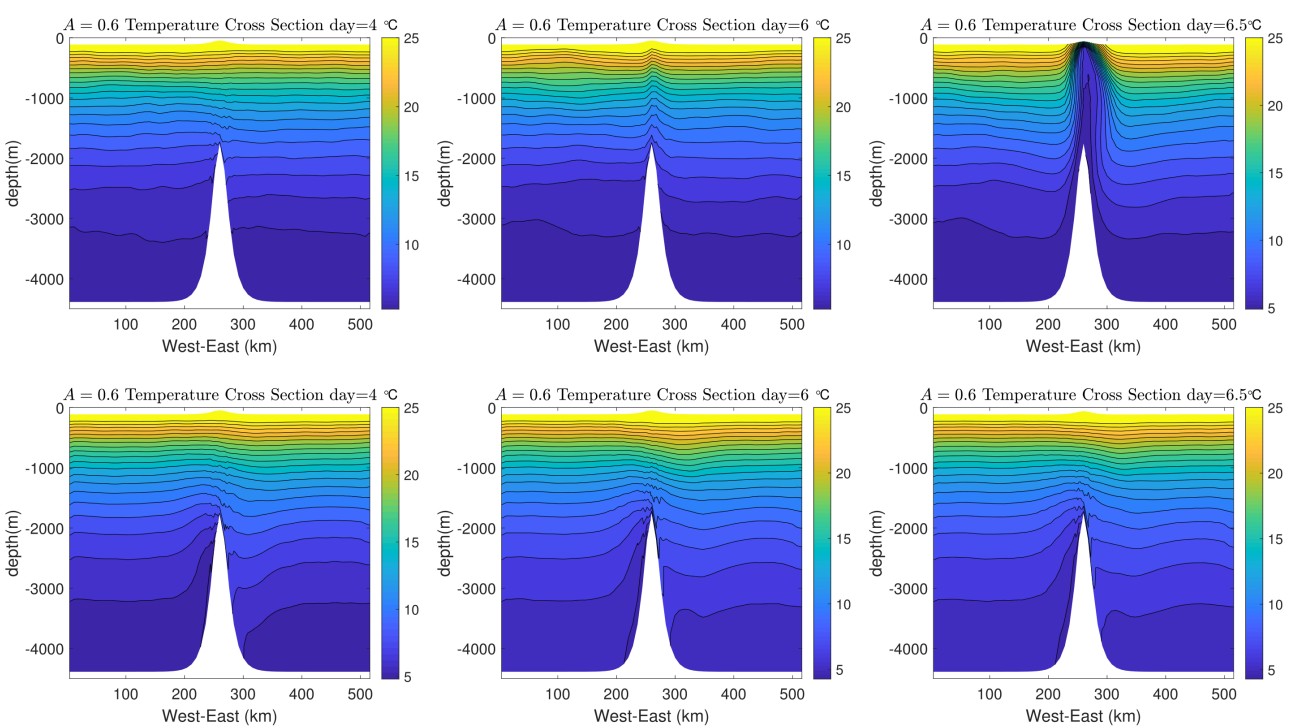

**Figure 13.** O-POM vertical temperature sections at $y = 196$ km under Coriolis-free(upper row) and Coriolis parameter $cor = 1.0 \times 10^{-4}$(lower row) at day 4 (left), day 6 (middle), and day 6.5 (right)

400    To investigate the destabilization mechanisms of O-POM scheme under zero Coriolis force, we conducted a case study with steep topographic parameter $A = 0.6$. Figure 13 displays vertical temperature sections at $y = 196km$ under Coriolis-free conditions (upper row) and Coriolis parameter $cor = 1.0 \times 10^{-4}$(lower row), showing model outputs at day 4 (left), day



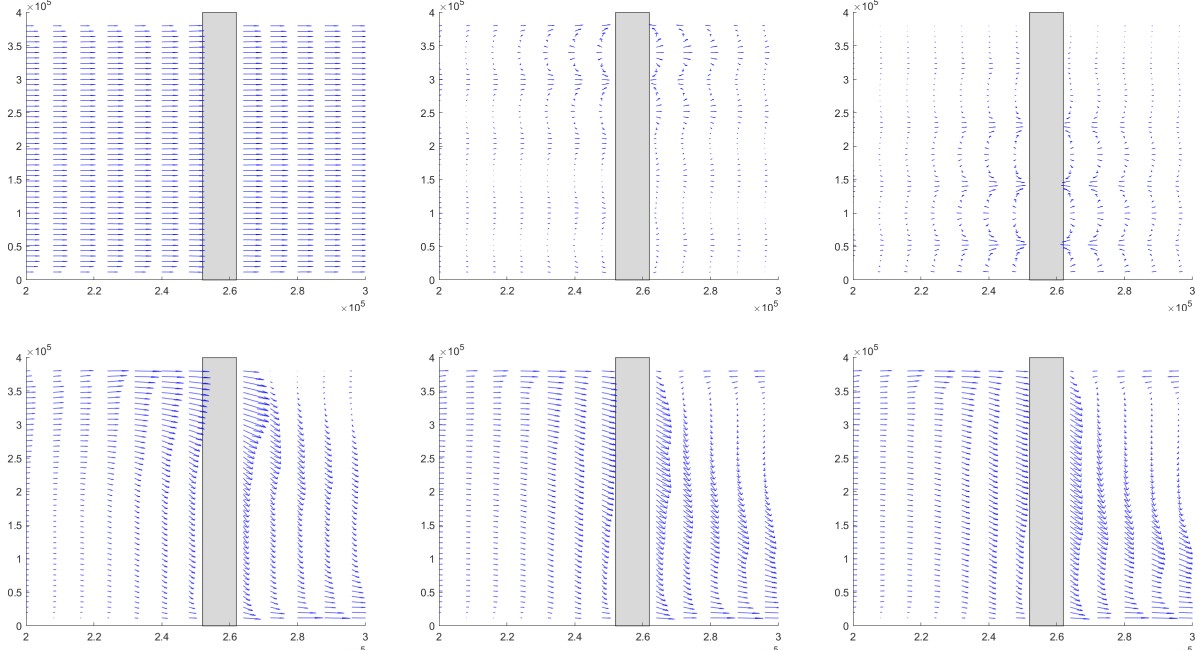

**Figure 14.** O-POM horizontal current fields at 2000 m depth under Coriolis-free(upper row) and Coriolis parameter $cor = 1.0 \times 10^{-4}$(lower row) at day 4 (left), day 6 (middle), and day 6.5 (right)

6 (middle), and day 6.5 (right). Correspondingly, Figure 14 presents horizontal current fields at $2000m$ depth for the same timestamps.

The experimental results demonstrate that under Coriolis-free conditions (upper row of Figure 13), the vertical temperature gradient intensifies progressively due to the absence of Coriolis-induced lateral deflection during sill overflow. This leads to persistent water accumulation at the sill crest, ultimately triggering numerical collapse. In contrast, under the Coriolis force condition (lower row of Figure 13), lateral deflection effectively reduces water accumulation intensity, enabling stable simulation of sill overflow processes.

Current field analysis (Figure 14) further reveals that without Coriolis force, significant flow reversal emerges by day 6, with dramatic increases in reversal velocity and spatial velocity gradient by day 6.5, directly causing model instability. With Coriolis force present, the current maintains an acute angle relative to the sill isobaths, accompanied by gradual velocity gradient variations that ensure computational stability.

Figure 15 presents simulation results under non-Coriolis conditions using the IMVL enhanced POM model, showing tem-
perature cross-sections (upper row) at y=196 km and horizontal velocity fields (upper row) at 2000 m depth for Day 4 (left), Day 6 (middle), and Day 6.5 (right). The IMVL scheme successfully captures stable seawater overflow dynamics across the topographic sill, as evidenced by diminishing temporal variations in thermal structure post-Day 6 and symmetric north-south distribution patterns in steady-state horizontal flow fields.



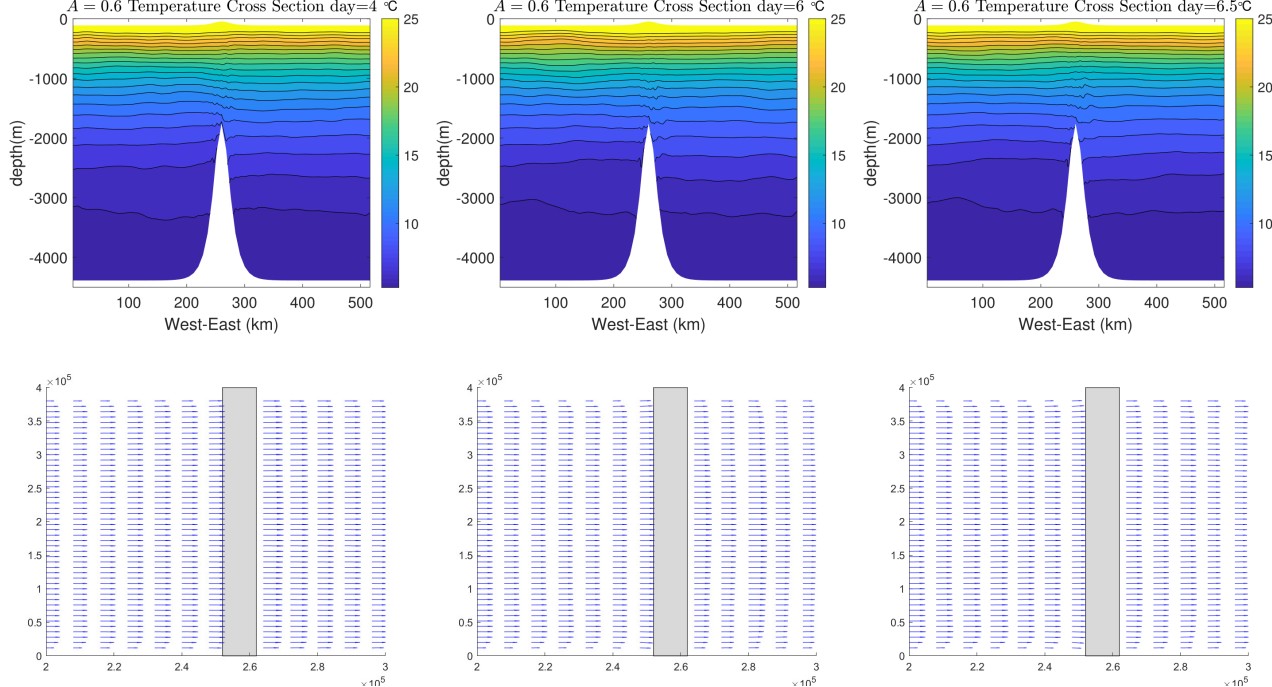

**Figure 15.** I-POM vertical temperature sections at $y = 196$ km (upper row) and horizontal current fields at 2000 m depth(upper row) under Coriolis-free at day 4 (left), day 6 (middle), and day 6.5 (right)

Theoretical analysis reveals that Coriolis force modifies flow trajectories to reduce effective topographic slope resistance. Specifically, Coriolis-induced flow deflection alleviates numerical instability caused by $\sigma$-coordinate transformation over significant bathymetric features. Crucially, the IMVL approach substantially enhances model stability in topographically complex regions, regardless of Coriolis force inclusion.

## 5 Conclusions

In this paper, the variable limit integral scheme is designed and constructed for the temperature and salt equation in the ocean motion governing equation under the Arakawa-C grid setting. According to the different characteristics of horizontal advection term, horizontal diffusion term and vertical diffusion term of the thermohaline equation, the corresponding variable limit integral scheme under C grid is designed and embedded in the POM ocean model. In addition, based on the convection equation, the stability of the variable limit integral scheme is analyzed separately.

Numerical experiments demonstrate that the implementation of the variable-limit integral scheme in the temperature-salinity equations significantly enhances the O-POM model's performance in terms of accuracy and stability. The core conclusions are as follows:



1. The Strait Zero-Startup numerical experiments demonstrate that, compared with the original Princeton Ocean Model (O-POM), the modified numerical model incorporating a variable limit integral scheme for solving temperature and salinity equations significantly reduces 40-60% simulation errors in seawater temperature and salinity.

2. The modified scheme exhibits heightened sensitivity to topographic variations, particularly in simulating overflow phenomena across steep sills, with computational discrepancies amplifying nonlinearly with both topographic steepness and simulation duration.

3. Computational efficiency analysis reveals that while the temperature-salinity module's calculation time triples, the overall computational overhead increases by merely 25%, remaining within acceptable thresholds.

4. Notably, in sill topography tests under zero Coriolis conditions, the modified formulation successfully resolves numerical instabilities that plague the original model after 6-8 simulation days. Physical mechanism analysis elucidates that traditional POM formulations generate unphysical water stacking in the absence of Coriolis forcing, leading to anomalous accumulation of vertical density gradients and horizontal velocity gradients. The variable-limit integral scheme effectively mitigates these numerical artifacts through improved discretization, achieving stationary flow simulations
that better align with physical principles.

The findings demonstrate that it should be emphasized that the variable-limit integral method developed for thermohaline equations represents an innovative and robust numerical scheme, which, while implemented in POM for validation, is directly transferable to other ocean modeling frameworks. Incorporating this method into ocean numerical models exhibits notable advantages in enhancing model stability and resolving dynamic processes over complex topography, providing an effective
approach for optimizing the dynamical framework of ocean models.

*Code and data availability.*  All codes and data files related to this paper is available at the link https://doi.org/10.5281/zenodo.16751812 (Li, 2025). This link includes Fortran code (Fortran code.zip), generated NetCDF data files, and Matlab code (Matlab code.zip). Executing the Fortran code generates NetCDF data files containing model outputs such as temperature, salinity, currents, bathymetry of grid cells, and sea surface height. The Matlab code is used to generate the figures presented in the paper.

*Author contributions.*  X. Li: Conceptualization, Methodology (derivation of variable-limit integration), Software (code implementation and modification), Validation (test case design), Visualization (plotting), Writing – Original Draft.

Z. Song: Conceptualization, Validation (test case design and result verification), Writing – Review & Editing (structural guidance).

X. Zheng: Validation (test case design), Writing – Review & Editing.

F. Qiao: Conceptualization, Methodology (derivation of variable-limit integration), Validation (result verification and test case design),
Software (code implementation and modification), Writing – Review & Editing (manuscript revision).

H. Zhou: Visualization (partial plotting), Writing – Review & Editing (English and Chinese proofreading).



M. Ji: Validation (result verification), Writing – Review & Editing (proofreading).

*Competing interests.* Authors have no competing interests to declare.

*Acknowledgements.* This paper was completed under the joint support of the Oceanic Institute of China and the School of Mathematical
Sciences of Harbin Engineering.





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
