# Peer review of "The Applicability of the Integral Method with Variable Limit in Solving the Governing Equations for Temperature and Salinity in an Ocean Circulation Model"

_EGUsphere, 2025_

## Referee Comment (RC1)

**A Review of**
**"The applicability of the Integral Method with Variable Limit in Solving the Governing Equations for Temperature and Salinity in an Ocean Circulation Model"**
**by F. Qiao et al.**
* * *
This study introduces the use of a method referred to as IMVL (Integral Method with Variable Limit) for the discretization of horizontal advection as well as horizontal and vertical diffusion of active tracers within ocean models. Section 2 is devoted to presenting the temporal discretization of the temperature and salinity equations in the POM model (a standard Leapfrog scheme with a Robert-Asselin filter) and to introducing a spatial discretization method referred to as IMVL. In Section 3, IMVL is successively applied to the discretization of 1D horizontal advection, horizontal diffusion, and vertical diffusion. The proposed discretization method systematically requires the inversion of a tridiagonal matrix to estimate the corresponding terms. At the end of this section, the numerical stability of IMVL for linear 1D advection is analyzed considering a Leapfrog-type time discretization. The proposed discretization schemes have been implemented in the POM model. In section 5, they are evaluated for a set of idealized test cases, in comparison with the schemes usually employed in POM (although their exact nature is not clearly specified). Based on these numerical simulations, the authors conclude that IMVL provides superior results in terms of accuracy and stability compared to the standard discretization used in the POM model.

In addition to containing fundamental issues, which I highlight below, the level of clarity, rigor, and pedagogical quality of this manuscript is well below the standards required for publication in GMD. Significant portions of the manuscript are difficult (if not impossible) to follow because of a combination of undefined notations and erroneous equations. The study is limited by an insufficient consideration of the overall formulation of ocean models and by a vague mathematical presentation of the proposed method (there is an incalculable number of equations that are wrong). I now outline the arguments supporting my recommendation against the publication of this study.

**Requirements before manuscript submission for publication**  It may be worth reminding the authors that reviewers are not expected to correct typographical errors in the manuscript, particularly when such errors lead to equations that are nonsensical. In a manuscript with seven co-authors, one would reasonably expect such errors to have been identified before submission. This remark also applies to the presentation of the paper, in which many notations are introduced without any explanation. The two central equations of the study (Equations (1) and (3)) appear to contain significant errors concerning the variables with respect to which differentiation is carried out. Furthermore, the title and text of the manuscript discuss the *integral method with variable limits*, whereas the terminology used in the cited references is 'Integral Method with Variational Limit'. Considering that this discretization method is a niche approach used in only a very limited number of publications, it further contributes to the confusion. In addition to the fact

that the reference paper Luo et al. (2017) supposedly introducing IMVL is not available online as open access. I regret to say that Section 2.3 appears to be either a compilation of excerpts from unrelated studies or possibly generated by an inadequately trained AI. This contributes to making the paper extremely poor from a pedagogical standpoint.

I now present my scientific arguments:

- **Discrete local and global conservation :** PDE solvers, including but not limited to oceanic models, are formulated by discretizing the flux-divergence (conservative) form of the governing equations, thus ensuring discrete local and global conservation of the relevant physical quantities. In this study, instead of approximating fluxes at the interfaces, the operators are directly evaluated at the cell centers, as in a conventional finite difference scheme. Specifically, $\partial_x(DUT)$ is discretized at the cell centers rather than $DUT$ at the interfaces. The same approach is applied to the horizontal diffusion operator, where $\partial_x(HA_H)\partial_x T + (HA_H)\partial_x^2 T$ is discretized at the cell centers, rather than discretizing $HA_H\partial_x T$ at the interfaces. The consequence of this is that, although the proposed numerical scheme may still approximate the correct continuous operator, the conservation properties of the continuous problem are lost at the discrete level. Such a lack of conservation is unacceptable for realistic simulations over long timescales.

- **Constancy preservation:** as emphasized by Shchepetkin and McWilliams (2005), an important constraint in the design of an ocean model is the constancy preservation for tracers (i.e., if the tracer is initialized as a spatially uniform field, it stays uniform regardless of the velocity field). Constancy preservation is essential to ensure that the compressibility or divergence of the flow will not change the scalar quantity that is advected. To satisfy this constraint, there is a consistency required at a discrete level between the continuity and the tracer equation. Setting $T \equiv 1$ in the temperature advection operator should provide an equation discretely equivalent to the continuity equation. First, the manuscript does not address how the continuity equation is discretized in POM. Second, I am skeptical that the discretization of the continuity equation has been appropriately adapted between O-POM and I-POM. In I-POM, a discretization scheme equivalent to (14) but for $e = D$ should be used to provide $\partial_x(Du)$ in the continuity equation. Constancy preservation is a fundamentally essential property for any ocean model.

- **Equivalence with compact schemes:** even if the scientific procedure used to obtain them is obscure, the discrete schemes you obtain appear to me to be identical to classical fourth-order compact schemes, which have been known for years. If you open Lele (1992) and consider his equation (2.1), applying the constraints to achieve fourth-order accuracy, you obtain (for $p = \partial_x e$):

$$\frac{1}{6}p_{i-1} + \frac{2}{3}p_i + \frac{1}{6}p_{i+1} = \frac{1}{2\Delta x}\left(e_{i+1} - e_{i-1}\right)$$

which is your equation (17). Then, considering equation (2.2) from Lele (1992) and applying the constraints to achieve fourth-order accuracy, you obtain (for $q = \partial_x^2 e$):

$$\frac{1}{12}q_{i-1} + \frac{10}{12}q_i + \frac{1}{12}q_{i+1} = \frac{1}{\Delta x^2}\left(e_{i+1} - 2e_i + e_{i-1}\right)$$

which is your equation (20) (Setting aside the fact that your equation is wrong, as it should be $q$ rather than $p$, $\delta_2(e_i)$ is also wrong $2e_i \rightarrow -2e_i$). Therefore, this study does not demonstrate the benefit of the IMVL method compared to standard compact schemes. For a reason unknown to me, the authors do not consider the discretization of vertical advection. It is worth noting that compact schemes (sometimes interpreted as a reconstruction with parabolic splines) have been used for years in the context of ocean models (e.g., within the ROMS, CROCO, and NEMO models), see Shchepetkin (2015) (eqn 2.5), or

Lemarié et al. (2015) (e.g. their App. B). Except that in this context, the compact schemes are used to approximate the interface fluxes rather than the cell-centered operators. Unless explained in a scientifically irrefutable manner, the relevance of the IMVL discretization is far from obvious. It would be interesting to elaborate on how this method differs from compact schemes that have been available for several decades (seminal work by Orszag and Israeli in 1974 and Lele (1992) has nearly 6000 citations).

- **Stability and multidimensional aspects:** Regarding the stability analysis presented in Section 3.4, it ignores the Asselin filter, which affects stability. Moreover, the result you obtain has already been given in Lemarié et al. (2015) (see their Table 2) in the more complete case that includes an Asselin filter. The present analysis yields a CFL number of $1/\sqrt{3} = 0.57735$, and when including an Asselin filter (with $\alpha = 0.05$) Lemarié et al. (2015) find a CFL number around $0.522$. Moreover, the impact of multidimensional aspects on the stability of advection is not addressed. It would have been interesting to verify that applying your scheme in 2D independently in the $x$ and $y$ directions leads to a stable scheme when a Leapfrog method is used.

- **Practical implementation:** A main reason why compact schemes are not used in the horizontal direction in ocean models is the need to solve a tridiagonal problem in a direction where the grid is decomposed into MPI subdomains, and where the presence of coastlines (and the associated boundary conditions) complicates the structure of the matrix to be inverted. This issue is completely overlooked in this study, in addition to the fact that horizontal resolution is generally not constant in models, which would require introducing scaling factors to account for this in the proposed discretization scheme.

*Detailed comments*

- "To address limitations in traditional discretization methods for ocean numerical modeling", what are those limitations?
- Equations (1) and (3): the errors are obvious at first glance.
- Equation (1): $R$ is not defined
- l. 118: U and V should not be interpreted as "flow rates", these quantities should be expressed as velocities in m s$^{-1}$; otherwise, Equations (1) and (3) would not be dimensionally consistent.
- l. 124: what is meant by "analytical treatment" ? The remainder of the paper relies on numerical methods; no analytical treatment is provided.
- l. 130: the classical Arakawa grids are defined in the horizontal plane. It is misleading to refer to the vertical coordinates as an 'Arakawa C-grid'; this is not meaningful. The staggering in the vertical is either referred to as Lorenz grid or Charney-Phillips grid.
- Fig. 1 : "POM brochure" ?
- Fig. 1 : $Z(k)$, $ZZ(K)$ and $Z(K + 1)$ are not defined
- Fig. 1 : $K_M$ is not defined
- The sentence "The first stage (...) for the horizontal advection and diffusion terms" is inconsistent with Equation (5), which also accounts for vertical advection. . .
- Equation (6): same error as in equation (1) and $R$ is still not defined
- l. 140: "format" ? The paper alternates between the terms 'format' and 'equation'; it is unclear whether 'format' is standard terminology.
- l. 142-144: the sentence "The POM ocean model uses a weak filter (Asselin, 1972) to filter out the computational modes brought by the time-splitting algorithm" is highly imprecise. Firstly, a Leapfrog-type scheme exhibits only a single computational mode. The nonlinear terms introduce a coupling between the physical and computational modes which may amplify the computational mode. This is the reason why filtering is necessary; it has nothing to do with the time-splitting (it is not even clear which time-splitting method is being referred to here. . . ). Finally, there is no justification for assuming

that this filtering is necessarily 'weak'.

- Sec. 2.3: This section is intended to be central to the rest of the study, yet it is practically incomprehensible, as none of the introduced notations are defined. It is not even possible to understand what the authors' objective is, or what $f$, $a_k$, and $b_k$ represent. The notations are ambiguous and nonstandard (what does $\int_{xx}$ mean?).
- Lemma 1: how does this lemma provide an approximation method? What is $K$ ? What is $R$ (certainly not the same as in Equation (1)) !
- The three equations between line 170 and line 175 are dimensionally inconsistent ($\Delta x^3$ and $\Delta x^2$ do not match the dimensions of the left-hand sides of the corresponding equations, i.e. $e(x)$, $p(x)$ and $u(x)$).
- The equation $u(x) = \ldots$ for the 2-point Lagrange interpolation is incorrect, it shoud be

$$u(x) = \frac{1}{\Delta x}\left(u_{i+1}\left(x - x_i + \frac{\Delta x}{2}\right) - u_i\left(x - x_i - \frac{\Delta x}{2}\right)\right)$$

  such that $u(x_i + \Delta x/2) = u_{i+1}$ and $u(x_i - \Delta x/2) = u_i$ (which is consistent with figure 2) and the $\Delta x^2$ has nothing to do in this equation.
- Equation (12) is incorrect, $\Delta x^2$ does not have the same units as the other terms.
- Remark 1 is correct, but it would have been helpful to note that solving a tridiagonal system is required, which is challenging in the horizontal direction because of the MPI domain decomposition employed by all ocean models. For this reason, compact schemes are never applied horizontally in these models.
- First equation of section 3.2: what is the rationale for rewriting this operator from divergence form into non-divergence form? This kind of thing should be avoided at all costs, as it breaks the global conservation properties of the quantity being diffused.
- Equation (18): you define $q$ but you never use it !
- Equation (20): again this equation is incorrect, the left hand side and the right hand side do not have the same units !
- Définition of $\alpha_0$ following equation (24): $d_z$ is not defined.
- l. 266: "the other parts continue to follow the POM model?", clarifying the numerical options of O-POM would have been valuable, because, as presented, it is not even clear which reference your schemes are being compared to.

**References**

Lele, S.K., 1992. Compact finite difference schemes with spectral-like resolution. J. Comp. Phys. 103(1), 16–42.

Lemarié, F., Debreu, L., Madec, G., Demange, J., Molines, J., Honnorat, M., 2015. Stability constraints for oceanic numerical models: implications for the formulation of time and space discretizations. Ocean Modell. 92, 124 – 148.

Luo, Y., Li, X., Guo, C., 2017. Fourth-order compact and energy conservative scheme for solving nonlinear klein-gordon equation. Numerical Methods for Partial Differential Equations 33(4), 1283–1304.

Shchepetkin, A., McWilliams, J., 2005. The regional oceanic modeling system (roms): a split-explicit, free-surface, topography-following- coordinate ocean model. Ocean Modell. 9, 347–404.

Shchepetkin, A.F., 2015. An adaptive, courant-number-dependent implicit scheme for vertical advection in oceanic modeling. Ocean Modell. 91, 38–69.

---

## Referee Comment (RC2)

Review of "The Applicability of the Integral Method with Variable Limit in Solving the Governing Equations for Temperature and Salinity in an Ocean Circulation Model"

This paper applies the IMVL method to the ocean tracer advection-diffusion equation, and implements it in POM. It is useful for ocean modelers to see new discretization methods tested in ocean settings. The authors provide sufficient details of the formulation for it to be reproduced by others. I think this paper is novel and interesting, and the topic is appropriate for publication in GMD.

My main criticism is that convergence tests against exact solutions are required for new methods of spatial discretization. This will verify the order of the method, and needs to be conducted for individual operators and for the full equation with time-stepping. See details in point 1. This will add a new section to the paper, and in my view is a requirement for publication.

Major items
1. A major item missing from this paper is convergence plots with rates of convergence for each operator and for the whole tracer equation together with time-stepping. This is required and basic for the introduction of all new numerical methods. Luckily there are numerous examples of such verification tests in the literature for advection-diffusion operators. See Bishnu et al 2024 for operator convergence tests and the method of manufactured solutions. This paper is for the shallow water equations, but includes examples in the supplementary material for advection-diffusion equations. The most elementary test for advection is a Gaussian distribution of tracer which advects back to the initial condition (the exact solution), on both a periodic plane and a sphere. See Section 2a in Skamarock and. Gassmann, 2011. Likewise, it is elementary to construct an exact solution for just a diffusion equation in time to measure the order of convergence of your method.
2. If the paper is too long with the inclusion of the new material recommended in point 1, some of th plots in the later sections could be moved to supplementary material or removed without loss of important content for the paper. For example, Fig. 6 and 7 could be merged and just one time slice chosen.
3. The authors appear to be careless in their writing, with simple mistakes such as wrong variables and missing spaces. See small comments below. I ask the authors to proofread their papers more carefully in the future.
4. I find equation 8 very non-intuitive. The IMLV idea seems to be based on this definition as a clever way to solve integral equations with specific, highly-accurate stencils. I can appreciate that, but it would be nice to have some physical intuition about equation 8. I read the previous two papers by Luo to understand the methods in this paper. Those also do not provide any explanation about the direction or reasoning behind these derivations, which was frustrating to me. For example, equation 8 integrates in x three times, but the outer two integrals are about the limits of the inner-most integral. So the units are [f x^3]. I suspect that there is a nice way to represent this spatially in a diagrammatic figure as a triple integral between epsilon_1 and epsilon_2, which would be nice to add to your paper. If the authors have some interpretation of eqn 8 that lends

some spatial or physical intuition to this method, please add it. If not, then explain that this is simply a mathematical formulation that allows us to solve the tracer equation with integrals rather than derivatives.

5. On equation (8), some readers are more used to the integral sign (integrand) dx notation with the dx at the end. In this case, that would put the dx dx_a dx_b at the end. You don't need to change it, as your notation is also used in the literature and follows the Luo papers. However, it would be good to comment at line 155 that (8) is a triple integral of f(x) in dx dx_a dx_b.

6. Under section 2, the authors need to provide context on the full use of the model for this method. Are the numerical methods for the solution of the momentum equation left unchanged? Are the layer thicknesses fixed (z-level), and what is the surface (rigid lid or free surface?). How do these equations interact with the new methods introduced in this paper? Why did the authors decide to only alter the tracer equations and not momentum?

7. Similarly, in the conclusion please discuss the potential for this method to be used for momentum and thickness/free surface equations. In the conclusion, please conjecture if the IMVL is appropriate for other types of models (B-Grid, unstructured, finite volume, sigma-coordinate, etc) or if there are any fundamental limitations that require that IMVL only be used on B-grid finite difference models.

8. Please comment in the conclusion if these methods may be used on a spherical domain in the current I-POM formulation. If alterations are needed, what are they?

Small items
Author's name Li appears in lower case "l" on author list.
3 a integral -> an integral (check throughout)
30 The Parallel Ocean Program (POP, Maltrud and McClean 2005) is a major finite difference ocean model missing from the list.
40 ICON (Korn 2019, Korn et al 2022) and MPAS-Ocean (Ringer et al 2913, Petersen et al 2019) are major finite volume codes missing from the review. They are unstructured grid models, like FVCOM.
eqn 1 first d/dT should be time, d/dt. vertical diffusion derivative shown at dT/dT should be dT/d\sigma.
eqn 4: same comment as eqn 1 but for S.
119 watch for spacing after commas: Similarly,the
160 scriptkis -> subscript k is (missing spaces)
295 Settings -> settings
342 should be: In Figures 6 and 7, the
Fig 6 and 7, captions should say what the rows are.
361 should be: Figures 8, 9, and 10 demonstrate that (spaces are missing)
Fig 10. Missing spaces in caption.

**References**

Bishnu, S., Petersen, M. R., Quaife, B., & Schoonover, J. (2024). A verification suite of test cases for the barotropic solver of ocean models. Journal of Advances in Modeling Earth Systems, 16, e2022MS003545. https://doi.org/10.1029/2022MS003545

Korn, P. (2018): A structure-preserving discretization of ocean parametrizations on unstructured grids. Ocean Modelling, 132, 73-90.

Korn, P., and Coauthors (2022): ICON-O: The Ocean Component of the ICON Earth System Model-Global Simulation Characteristics and Local Telescoping Capability. J Adv Model Earth Syst, 14, e2021MS002952, https://doi.org/10.1029/2021MS002952.

Mathew E Maltrud, Julie L McClean,
An eddy resolving global 1/10° ocean simulation,
Ocean Modelling,
Volume 8, Issues 1–2,
2005,
Pages 31-54,
ISSN 1463-5003,
https://doi.org/10.1016/j.ocemod.2003.12.001.
(https://www.sciencedirect.com/science/article/pii/S1463500303000684)

Petersen, M. R., Asay-Davis, X. S., Berres, A. S., Chen, Q., Feige, N., Hoffman, M. J., et al. (2019). An evaluation of the ocean and sea ice climate of E3SM using MPAS and interannual CORE-II forcing. Journal of Advances in Modeling Earth Systems, 11, 1438– 1458. https://doi.org/10.1029/2018MS001373

Ringler, T., Petersen, M., Higdon, R.L., Jacobsen, D., Jones, P.W., Maltrud, M., 2013. A multi-resolution approach to global ocean modeling. Ocean Modelling 69, 211-232. https://doi.org/10.1016/j.ocemod.2013.04.010

Skamarock, W.C. and A. Gassmann, 2011: Conservative Transport Schemes for Spherical Geodesic Grids: High-Order Flux Operators for ODE-Based Time Integration. Mon. Wea. Rev., 139, 2962–2975, https://doi.org/10.1175/MWR-D-10-05056.1